# ColEval: Honeybee COLony Structure EVALuation for Field Surveys

**DOI:** 10.3390/insects11010041

**Published:** 2020-01-05

**Authors:** Julie Hernandez, Alban Maisonnasse, Marianne Cousin, Constance Beri, Corentin Le Quintrec, Anthony Bouetard, David Castex, Damien Decante, Eloïs Servel, Gerald Buchwalder, François Brunet, Estelle Feschet-Destrella, Kiliana de Bellescize, Guillaume Kairo, Léa Frontero, Miren Pédehontaa-Hiaa, Robin Buisson, Theo Pouderoux, Alexandre Aebi, André Kretzschmar

**Affiliations:** 1Laboratory of Soil Biodiversity, Institute of Biology, University of Neuchâtel, 2000 Neuchâtel, Switzerland; 2ADAPI, Maison des Agriculteurs, 22 Ave H. Pontier, 13626 Aix en Provence, France; a.maisonnasse.adapi@free.fr (A.M.);; 3UMT PRADE, INRA, Site Agroparc, 84914 Avignon, France; 4INRA, Unité Abeilles & Environnement, Site Agroparc, 84914 Avignon, France; 5ITSAP, INRA, Site Agroparc, 84914 Avignon, France; 6ADA-Occitanie, 2 rue D. Brisebois, BP 82256, 31322 Castanet Tolosan CEDEX, France; 7Fondation Rurale Interjurassienne (FRI), 2852 Courtemelon, Switzerland; 8Professional Beekeeper, Miels Brunet, 1315 La Sarraz, Switzerland; 9ADAPIC, Cité de l’Agriculture, 13 Avenue des Droits de l’Homme, 45921 Orléans CEDEX, France; 10ADANA, Maison de l’Agriculture, Cité Galliane, 55 Avenue Cronstadt, 40000 Mont-de-Marsan, France; 11ADA AURA, Chambre Régionale d’Agriculture, 9 allée Pierre de Fermat, 63 170 Aubière, France; 12Laboratory of Soil Biodiversity, Institute of Ethnology, University of Neuchâtel, 2000 Neuchâtel, Switzerland; alexandre.aebi@unine.ch; 13INRA, Unité Biostatistique et Processus Spatiaux, Site Agroparc, 84914 Avignon, France; andre.kretzschmar@inra.fr

**Keywords:** honeybees, colony structure, field experiment, human estimations of colony hive

## Abstract

Methods for the evaluation and comparison of the structure of numerous honeybee colonies are needed for the development of applied and fundamental field research, as well as to evaluate how the structure and activity of honeybee colonies evolve over time. ColEval complements existing methods, as it uses an online reference image bank for (human) learning and training purposes. ColEval is based on the evaluation of the surface area percentage occupied by different components of a honeybee colony: adult worker bees, open and capped brood, honey, nectar, and pollen. This method is an essential tool for the description of the evolution in the size of honeybee colonies. The procedure makes allowances for tendencies between different observers and uses them to calculate accurate measurements of honeybee colony evaluation. ColEval thus allows for a posteriori comparison of under- or over-evaluation made by different observers working on the same project; it is thus possible to eliminate observer bias in the measurements and to conduct large surveys.

## 1. Introduction

In order to provide the practical knowledge that will help beekeepers to maintain colonies; secure honey productivity; minimize damage from bio-stressors (*V. destructor* [1,2], virus, *Nosema* spp., …); and apprehend environmental constraints such as habitat loss, flower paucity, and pesticide exposure, honeybee researchers need precise and efficient tools that deliver comparable data [3]. In the current context of worldwide concerns about honeybee decline [4,5,6,7,8], field research at an apiary scale requires effective and efficient methods for the evaluation of the population structure of colonies [9,10]. The key components of the colony structure are the number of adult worker bees and the number of capped and open brood cells, which together indicate the future size of the colony. Food stores necessary for the colony (honey, nectar, and pollen stored in combs) are crucial to ensure survival during autumn and winter periods when foraging is not possible. Thus, the presence of such stores is a good indicator of colony vitality and resilience.

Several methods already exist to evaluate the different components of honeybee colonies occupying hives [11,12,13], using an assessment of colony components on combs with a partition of the observed space into a small number of quadrats of equal area. If the number of quadrats is small (between 6 and 9, for example), the measure of several colonies is better fitted by a multinomial variable. The higher the number of quadrats, the closer the counting is from the continuous variable, which is more suitable for downstream statistical analysis and modelling [10,14]. Other methods use combs photography to allow for reliable visual assessments, but quality photographs are not easy to obtain under field conditions (variable weather, honeybee behavior, and so on). Even if the counting can be done by image analysis in silico, the result of automatic image analysis is hampered by limitations in image definition (low quality of contrast that could lead to an error of counting evaluation) [15,16].

Improvements in honeycomb image analysis are underway [17,18,19,20,21,22]; the use of these techniques is impaired by the time needed to take and analyze photographs. These methods do not yet offer a suitable way to describe a large number of colonies per day or, consequently, to record the high variability attached to hives and apiaries. The structure of honeybee colonies has been shown to be highly variable in large surveys at apiary scale, or even during one particular honey flow at colony scale [23,24]. For this reason, it is important to consider the number and the variability of colonies observed, because it has a crucial bearing on the statistical validity of most experiments. For example, cohorts of a minimum of 25 hives per group compared per replicate are needed when attempting to validate the efficiency of *V. destructor* treatments, or when providing specific instructions for the use of new products or methods against mites (Kretzschmar, unpublished data). The development of image analysis methods has not yet been used in large apiary surveys, which is most likely related to the time cost associated with these techniques.

A high number of colonies observed repeatedly over time, with minimal disturbance to the study objects, is also required by recent approaches addressing the modelling of colony structure dynamics [25], which, again, require a rapid method of colony size estimation. A large team is needed to observe multiple colonies, and it is thus often necessary to involve people with varying levels of expertise (beekeepers, engineers, technical staff, and researchers) in the research projects to evaluate the colonies. Individual variations in assessments between observers must thus be taken into account in order to deliver standardized and comparable measures. The common bias encountered in social sciences; that is, inexperienced participants, false counting, observer expertise evolution [26,27], should be addressed by developing learning, training and evaluation support for all team members, regardless of their level of expertise.

It is for these reasons that we chose to improve an easy to handle field technique, supported by evaluation tools and fitting the statistical and modelling needs in honeybee epidemiological research [24,28,29,30,31,32,33]. In the long-term surveys of honeybee weight performance under lavender or sunflower honey flow [24,30], 200–600 hives are surveyed at the beginning and at the end of honey flow (circa 2000 to 6000 frames evaluation). The development of a large and long-term survey is the first step in honeybee epidemiologic studies. ColEval is devoted to being used for this development. Image analysis would provide reference counts of open or capped brood cells, but it fails in giving accurate counts of bees. ColEval is presented here as a convenient trade-off between accuracy and efficiency in describing, under field conditions, the components of a hive at a given time.

In order to account for the aforementioned constrains, an efficient and precise description method with human learning and comprehensive training is needed to (i) describe a sufficient number of apiaries for field experiments and (ii) measure the statistical variability between observers when experiment scale and schedule requires division of labor between several persons. The method presented in this paper offers a solution to the above-mentioned constraints, specifically addressing the aims of being able to describe a large number of honeybee colonies while maintaining statistical validity. To ensure the highest quality of data when quantitatively evaluating the five components of colony structure (number of adult workers, number of open and capped brood cells, surface area of honey, nectar, and pollen stores) and enable assessment of a large number of colonies per day, we created a set of three learning and training tools: (i) a learning, training, and self-evaluation tool; (ii) a specific field application tool to improve the quality of the counting; and (iii) a method to compare and adjust results from different observers. The impact of performing repeated ColEval measurements on honeybee colonies was also tested to validate this method.

Nevertheless, the authors make clear that ColEval is a method of evaluation that provides a training tool for minimizing the variability of the measures at individual level, knowing this method includes the inherent variability of human quantification.

## 2. Materials and Methods

### 2.1. Description of the ColEval Method

The ColEval method consists of the description of the entire colony; both faces of each comb of the brood box and suppers are screened, along with their walls, to visually estimate the quantity of adult workers. The observer analyzes every comb in the hive. Once the evaluation of a comb is complete, it is placed in a different box to prevent workers changing combs, as this would bias the assessment of the next combs. Once all combs have been screened, the observer evaluates the number of adult bees on the walls of the inspected hive.

Each component (adult workers, capped and open brood, honey, nectar, and pollen) is evaluated through the proportion of the surface it covers on each comb side and wall (the latter for adult workers only). This proportion is converted into a percentage of the total comb surface and is then transformed (see below Section 2.3 and Table 1) into the following:Number for worker bees;Number for capped and open brood cells;Area (dm^2^) for cells filled with honey/nectar or pollen.

For practical reasons, the percentages are rounded to the nearest 5 (counting by steps of 5 percentage points). When several layers of worker bees cover a comb, the estimation could be higher than 100%.

### 2.2. Learning and Training

• Image bank and reference counts

An image bank with photos (n = 300) of comb sides covered with various numbers of adult worker bees and photos (n = 600) of comb sides from which adult workers have been removed to make the capped brood cells visible [34]. Reference images for the open brood cells are not provided here, because they are particularly hard to photograph, even under laboratory conditions (43 h for 16 photos of high quality in special light chamber; P. Le-Bivic, INRA, personal communication), are and difficult to see on photos taken in apiary conditions owing to light conditions [12,19] (brood comb picture in “SM3_brood comb.JPG”). Adult workers and capped brood have been manually counted on these photos with the help of ImageJ software to provide a reference count [35]. Several formats of hives are used by beekeepers. The main formats in Europe and the United States are Dadant and Langstroth. To cope with these differences, combs of the latter two formats are included in the picture bank and a rule of transformation of percentages to counts is provided (Table 1). The image bank (photos of combs or virtual images), learning tools, and instructions for use are available via the permalink: http://w3.avignon.inra.fr/lavandes/biosp/colevalENG.html.

• Learning training and processes

During the learning and training sessions, several series of 20 photos drawn at random from the image bank are displayed on a computer screen to simulate a whole hive. Evaluation of either the number of adult bees or capped cells is typed in by the observer, recorded by the R software [36], and then compared to the reference counts. At the end of a counting series (20 images), several descriptors are calculated, as follows:

the average absolute error for each counting: $\frac{\sum_{1}^{n}\left|evaluation-reference\right|}{n}$, calculated from a series of *n* counts (see Figure 1);

the effective error of the total series: $\sum_{1}^{n}evaluation-\sum_{1}^{n}reference$, for a series of *n* counts;

the effective relative error for the total series: $\frac{\sum_{1}^{n}\left(evauation-reference\right)}{\sum_{1}^{n}reference}$.

Finally, for the regression coefficient α of the counting series (see Figure 2), the regression follows the line equation *Evaluation* = α. *Reference* (*Intercept* = 0), where α depicts the tendency of the observer to under or over-evaluate and provides the coefficient of correction when the counting of several observers is to be compared. The reference counting of each observer is then *Reference_observer i_ = Evaluation_observer i_/α_observer i_*.

Along successive series, the observer can thus evaluate her/his ability to describe the comb components based on observation of her/his own errors. For each photo, a reference count is provided. This enables the observer to see why their estimation was wrong; especially, the observer can check if the error depends on the quantity of adult bees.

• Additional corrective procedure for adult honeybee workers

We hypothesized that the assumption that honeybees formed a unique layer on the comb was in most cases not met in field realistic settings. In an attempt to better estimate the number of adult worker bees on a comb, we compared the estimates obtained with the ColEval method described above with an estimate obtained by weighing the workers. Five observers performed the assessments of 150 combs simultaneously. We considered that the weight of one individual equals 0.140 mg [37]. From this comparison, a coefficient of under- or over-evaluation, as well as the average coefficient, was estimated for each observer (Figure 3).

• Correction between observers

Three observers performed ColEval evaluations (Figure 4). Each of them described one-third of the same colonies in each apiary to determine inter-individual errors. The evaluation of each observer is corrected, for each component, taking the mean of all the counts as the reference and applying a correction coefficient (=*mean*(observer)/*mean*(total)) to each observer (Figure 4).

New observers are expected to train with an experienced observer. Thus far, all observers in the various programs in which ColEval is used started with the initial group who established the method (namely, Julie Hernandez, Alban Maisonnasse, and Andre Kretzschmar). Observers are expected to engage in regular training and to maintain their level of accuracy. Nevertheless, it is clear that specific trends (either over or under estimation) are attached to each observer. As it is always possible to account for these trends, the main point is to check that the trend is constant.

### 2.3. Data Collection Support Spreadsheet

Field spreadsheets used to document the coverage percentage of each of the five components on each side of each comb of the hive are provided in the Appendix A. They are transcribed on formatted spreadsheets (format type .xlsx, .csv, .ods, or others), which transform percentage evaluation into numbers or surface areas, with the coefficient transformation values given in Table 1 (see above in Section 3.1. Data transformation).

Calculation of the theoretical number of cells and adult workers on comb.

The number of cells covering 100% of a comb side was measured on 30 combs. For the evaluation of the number of adult honeybee workers covering 100% of a comb side, the body surface area of 130 individuals was measured with image analysis (ImageJ—[35]). Hypothesizing that honeybees formed a unique continuous layer, the maximum number of honeybees needed to cover 1% of a comb side was estimated (See Table 1). This maximum number fits nicely with the number obtain by Imdorf et al. [13] and Burgett and Burikam [38].

### 2.4. Impact on Colony

Two groups of 50 colonies were surveyed with ColEval at one-month and one-week intervals, respectively, during a four-week period. The first group was thus surveyed twice, whereas the second was surveyed five times.

## 3. Results

### 3.1. Data Transformation

To transform the percentage evaluation into numbers or surface areas, we calculated the coefficient transformation values given in Table 1. Each value given in the table is the number of items (adult workers or cells) or the comb area (dm^2^) corresponding to 1% of the total area (=area inside the wood comb) of a comb side of a given hive format.

The estimated percentage of the theorical number of cells covering the comb is transformed into number of cells (40 cells for Dadant hive type, 30.2 cells for Langstroth hive type, and 20 cells for Dadant supper represented 1% of the comb—Table 1). The estimated percentage of honeybees covering the comb is transformed into the number of worker bees (14 honeybees for Dadant hive type, 11 honeybees for Langstroth hive type, and 7 honeybees for Dadant supper represented 1% of covered comb—Table 1). One hundred percent coverage gives the maximum number of bees in one continuous layer on a comb.

### 3.2. Method Validation

• Learning process

The average absolute error for each counting and the regression coefficient α of the counting series (see above in Section 2.2 Learning training and processes) are presented in Figure 1 and Figure 2.

• Additional corrective procedure for adult honeybee workers

As seen in Figure 3, a coefficient of under- or over-evaluation as well as the average coefficient was estimated for each observer. The estimation of honeybee numbers by ColEval should be multiplied by a coefficient of 1.8 to more accurately approach the real number of worker bees. Additionally, the distribution of the residual errors both for each observer and on average for the five observers is presented in Figure 5.

• Standardization between observers

Taking observer tendency α on field measurements into account improves the quality of the results. Figure 5 illustrates the distribution of the evaluation of the five ColEval components by three observers, and their respective learning score. This figure clearly shows that observer #3 under-evaluated the components compared with the two other observers. To level off the evaluations of observer #3, they were corrected using the mean values for each component evaluated by the two others as reference (see above in Section 2.2. Correction between observers).

• Impact on colony

In our field experiments, no disturbance of honeybee colonies was noticed in terms of colony performance (weight gain) in short-term experiments. The average total weight of the colonies at the end of the survey was not different between the two groups (32.96 kg ± 4.6 and 33.5 kg ± 4.8, respectively).

## 4. Discussion

ColEval has already been applied to several large surveys (totaling approximately 15,000 colonies) within research programs of the research unit on honeybees at the National Institute of Agronomic Research (INRA) of Avignon (France), and many scientific publications used it in their monitoring design to characterize honeybee colony size [24,28,29,30,31,32,33]. Application of the method on this scale has clearly demonstrated that ColEval is an easy method to learn and to use. Two people (one observer and one person in charge of taking notes) can assess the demographic and food store components of about 30 to 40 colonies per day. As ColEval observers increase their skills with practice over time and with regular training sessions, the correction of observer bias can be better taken into account and can be used to better analyze the results. In fact, some observers have more difficulty estimating when adult bees are high in number, whereas for others, it is the contrary. In the case of capped brood evaluation, some observers struggle to take into account the presence of mosaic brood (Kretschmar unpublished). Repeated training sessions thus improve the quality of observers’ estimates.

To address this bias correction, it is advisable to record the identity of the observer on the field sheet when several observers are working on the same project (see Appendix A).

In our opinion, ColEval improves the methods developed by Liebefeld [10,39] and the methods based on combs photography [16,40], because of the increased precision deriving from its learning method on a broad photographic basis, and because of the possibility to compensate for over- or underestimation for each observer. This also allows for assessments made by several observers participating in a project to be homogenized/standardized. Compared with earlier methods, ColEval proposes a single way to measure all the components of a colony with percentage evaluation. ColEval yields a precise description. As the coverage percentage of all the elements (excluding adult worker bees) amounts to 100%, the observers can take an intuitive partition of the whole area of the comb surface into its components (including voids). Doing this minimizes the total error, which is averaged when the evaluation of the different components are summed up, while the errors of each component are summed up if the components are evaluated independently.

The ColEval method has shown (see Section 2.3) that the number given as reference for the number of worker bees on a comb surface by the Liebefeld method [39], or in methods described in the *BEEBOOK* [10], should be increased to improve the accuracy of the evaluation. The difference between these methods and ColEval could lie in the different behaviors of workers bees, which could be more intensely clustered together in our case. The differences in individual honeybee weight cannot be excluded either.

As the evaluation of the number of bees by weight is time demanding for field evaluation, it may be more efficient to evaluate a coefficient correction with a preliminary experiment, which can be repeated from time to time (as also mentioned in Imdorf et al. [39]). Then, instead of considering that 1400–1500 workers bees completely cover one Dadant comb face, it is more realistic to consider 2610 workers bees as the reference (1450 × 1.8 = 2610; see Figure 3). The same result is obtained by Burgett and Burikam [38].

This correction is also useful because it provides a more realistic quantification owing to the overlapping of adult worker bees that can be better interpreted by beekeepers participating in an apiary-level assessment of colony strength and dynamics. Additionally, applying this correction coefficient on the evaluations of the number of adult worker bees on 600 colonies nested in 24 apiaries of a large survey (unpublished data) shows an average colony size, at the time of this experiment (colonies on lavender honey flow in June in South France), of 17,514 ± 8321 worker bees. This number is closer to data found in literature from “Winston, 1987; Burgett and Burikam 1985” [38,41] compared with the estimation before correction of 9730 ± 4572 adult workers per colony.

Compared with the previous methods of describing colonies based on the evaluations of one or several observers, ColEval provides user-friendly training and learning tools required for the candidates to “calibrate” themselves before describing colonies. ColEval addresses this by providing a large image bank (adult workers and capped brood cells) as a tool for observer self-training without requiring handling and weighing of colonies.

Another problem with some other observation methods is that the time required for assessments could be incompatible with fieldwork conditions (variable weather, adverse weather conditions, too cold condition for brood, robbing, bee’s aggressiveness, and so on). In addition, these methods do not work for frequent and repeated description of the large number of colonies required by large surveys and would disturb the colonies for too long.

In our field experiments using the ColEval method, no disruption of honeybee colonies was noticed in terms of colony performance (weight gain) in short-term experiments. This result is compatible with those presented in the Liebefeld method with a three-week interval survey [39]. It still has to be proven that performing ColEval on colonies at one- or two-week intervals does not affect long-term honeybee behavior, especially regarding reproduction.

As ColEval provides continuous variables, it facilitates the use of these variables in generalized linear models, mixed or not. It enables better evaluation of the variability of these components in order to more accurately evaluate the response of population structures to bio-stressors and environmental conditions.

Despite the improvements brought by ColEval, it should be noted that the evaluations obtained must be considered as approximations. The way to make these approximations as realistic as possible is to practice and to perform regular training sessions using the image bank and reference counting.

## Figures and Tables

**Figure 1 insects-11-00041-f001:**
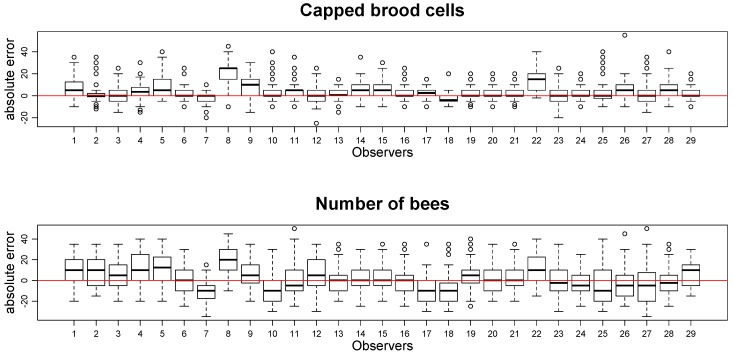
Absolute error of 29 learning observers for capped brood (upper) and worker bees (lower). Each observer has evaluated two or three series (20 images each) of both capped brood and honeybee images.

**Figure 2 insects-11-00041-f002:**
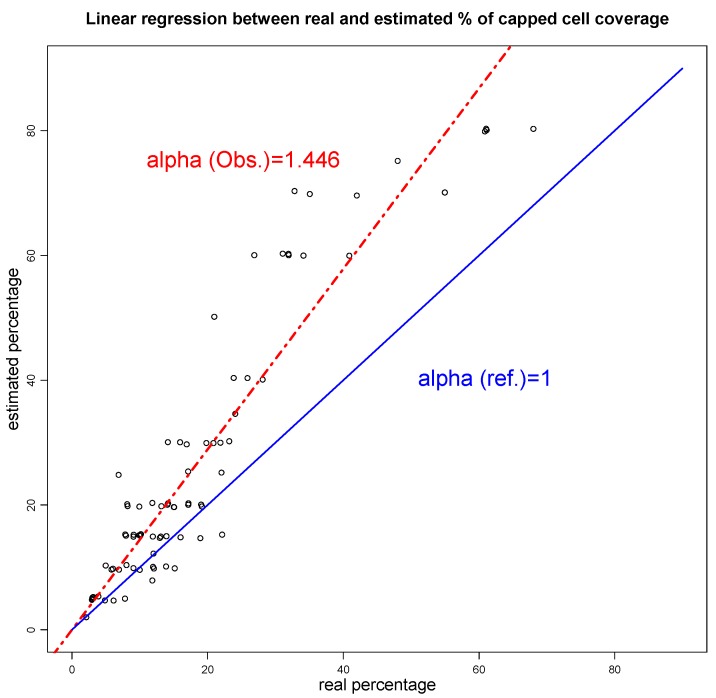
Linear regression between real and estimated percentage of capped brood coverage for two series of 20 images showing the tendency for one observer to deviate from the real counts: red dot-dashed line shows the observer evaluation; blue solid line is the reference line (α = 1). The coefficient α = 1.446 give the tendency of over-evaluation of this observer.

**Figure 3 insects-11-00041-f003:**
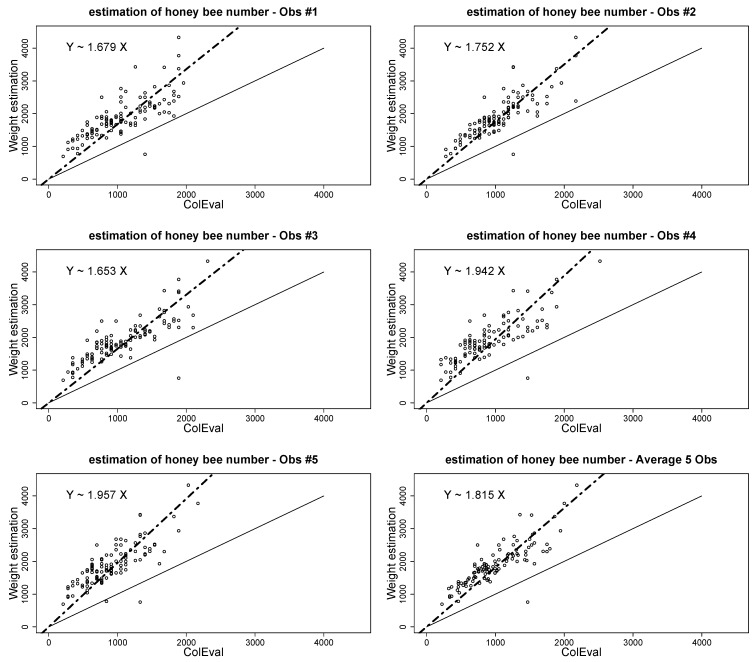
Linear regression between ColEval evaluation and weight-based evaluation of number of adult honeybee workers (dot dashed line). The equation Y = αX gives the tendency of over-evaluation; solid line is the line with α = 1. *X*-axis: bee number evaluated by ColEval; *Y*-axis: bee number evaluated by weighing.

**Figure 4 insects-11-00041-f004:**
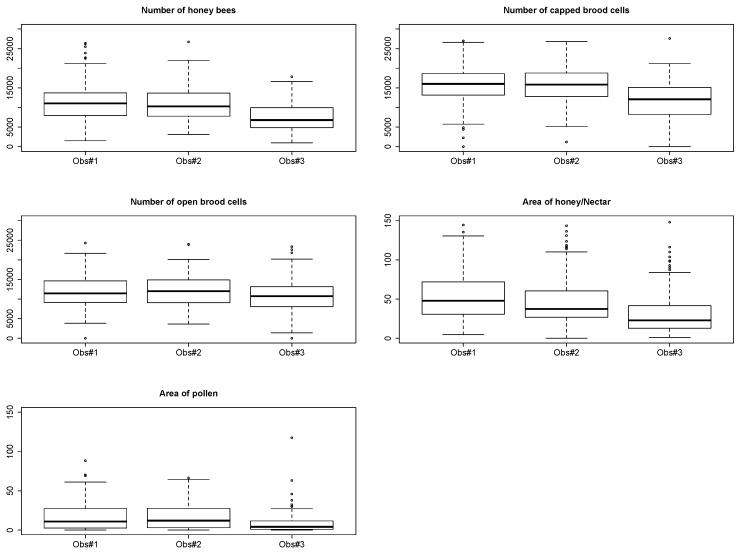
Comparison of the distribution of the evaluation of the five ColEval components by three observers.

**Figure 5 insects-11-00041-f005:**
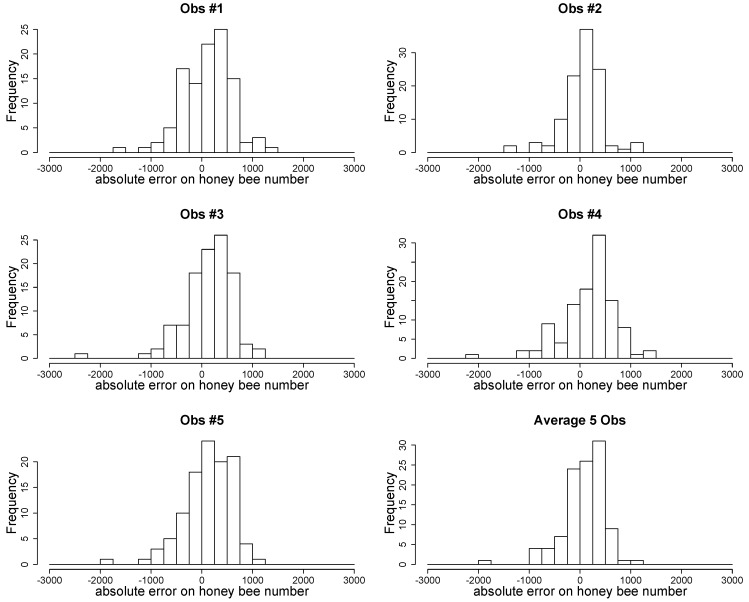
Histogram of residuals of the linear regression for the five observers and on average.

**Table 1 insects-11-00041-t001:** Estimated values of the coefficient to transform the percentage of comb coverage into number of brood cells and adult workers or into area of honey/nectar and pollen. Values given for 1%.

Hive Type	Number of Honeybees	Number of Capped Cells	Number of Open Cells	Area of Pollen (dm^2^)	Area of Honey/Nectar (dm^2^)
Dadant (hive)	14	40	40	0.1134	0.114
Langstroth (hive) *	11	30.2	30.2	0.0903	0.0903
Dadant (supper)	7	20	20	0.057	0.057

* For Langstroth hive, the suppers are generally either another Langstroth hives or Dandant suppers.

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
