# Peer review of "ColEval: Honeybee COLony Structure EVALuation for Field Surveys"

_insects, 2020, doi:10.3390/insects11010041_

Round 1

Reviewer 1 Report

Manuscript Title: “ColEval: Honeybee COLony structure EVALuation for field surveys”

Review Report:

This manuscript presented by Hernandez et al. describes a training method for observers with the potential to statistically alleviate human count errors while evaluating honey bee colony performance. The method also provides training for observers to improve the accuracy of their visual identification and estimation of major hive variables. An accurate evaluation of the colony performance such as brood production, pollen & honey cells, number of bees…etc is crucial to understand the effect of biotic and abiotic stressors on honey bee performance in the field. Therefore, this effort by Hernandez et al. has merit, and this proposed method could contribute in filling the error gab between the real count of a giving variable and the estimated count by the observer, BUT to a certain extent no proven yet in this MS. The authors also tackled the question of having multiple observers involved in the same project and the potential of curing each observer’s errors by calculating an alfa coefficient per observer. The manuscript is well written and presented, except some typos here and there to be fixed. Digital numbers should follow one standard (American or EU) across the text; I don’t know what type is the one presented in the text (ex: L267: 1’450, L274: 17’514…and so on). That said, I have some reservations related to the method presented here and particularly absence of citation to some more powerful automated and semi-automated tools available for this purpose and that could enrich the discussion and tool comparison in this MS.

This method will remain a relative quantification no matter how much or how long observer is trained. It should be made clear that an absolute quantification (or at least an accurate quantification) cannot be obtained using this method and observer should rely on alternative tools. Author should mention for readers the presence of fully automated tool to count capped brood, honey bees, honey cell, larva, pupae and even eggs. I believe German software were developed for this purpose few years ago: http://wsc-regexperts.com/en/honeybees/honeybee-brood-colony-assessment-software, I am however not sure if scientific papers were published covering these tools. Authors should discuss/mention also semi-automated system of brood counting, see please (Colin, Bruce et al. 2018).   I do not agree with the dismissal proposed or supported by authors of evaluating hive performance by taking photos and later on analyzing them by software. This method remains until now the most appropriate and accurate method to obtain a close to absolute count. It has been used in many studies (Abràmoff, Magalhães et al. 2004, Alburaki, Boutin et al. 2015, Alburaki, Steckel et al. 2017) and I agree that it is time consuming and the field conditions may not be optimal …harsh intervention on hive..etc but there should be a trade off here if one is willing to obtain real quantification of the hive performance.      Authors did compare inter-observers’ variations, but did not test intra-variation resulting from the same observer throughout time. In other words, I am confident that the coefficient of the same observer will eventually fluctuate by time and cannot be stabilized regardless of the number of training session undertaken. I did not see in this manuscript an evaluation of potential count change/error of the same observer throughout time; I do think that the factor time is crucial here. To validate this method, or at least the training tool, a proof of improvement (decrease in observation error count by training time) in the observer’s count after multiple training session must be provided. This is the only way to test the efficiency of this learning method as claimed by authors, unfortunately this data is missing in this MS. See L 265, time factor was mentioned but no proof of improvement by time was presented. L279: “Calibrate” themselves… this calibration cannot be sustainable and stable in any case, and how long will it last for? This bring me back to my point N4: a proof of this calibration and improvement in the observer count is needed, same for L298.                             

I would suggest to include the related references (or other similar refs) I cited above and broaden the discussion of this MS for a fair assessment of the different tools available in the market regarding colony performance evaluation.  

Finally, in my personal view, the observer will remain a human being and I do not think that learning tools would provide much of a difference statistically speaking, but rather focusing on improving algorithms, image filters and automated tools would be the answer to this challenge. Neutrality and accuracy tend to come by minimizing human intervention in such type of observation not the other way around.     

Abràmoff, M. D., P. J. Magalhães and S. J. Ram (2004). "Image processing with ImageJ." Biophotonics International 11(7): 36-42. Alburaki, M., S. Boutin, P. L. Mercier, Y. Loublier, M. Chagnon and N. Derome (2015). "Neonicotinoid-Coated Zea mays Seeds Indirectly Affect Honeybee Performance and Pathogen Susceptibility in Field Trials." PLoS One 10(5): e0125790. Alburaki, M., S. J. Steckel, M. T. Williams, J. A. Skinner, D. R. Tarpy, W. G. Meikle, J. Adamczyk and S. D. Stewart (2017). "Agricultural Landscape and Pesticide Effects on Honey Bee (Hymenoptera: Apidae) Biological Traits." J Econ Entomol 110(3): 835-847. Colin, T., J. Bruce, W. G. Meikle and A. B. Barron (2018). "The development of honey bee colonies assessed using a new semi-automated brood counting method: CombCount." PLoS One 13(10): e0205816.

Reviewer 2 Report

Dear Editor,

The manuscript addresses some important questions regarding the respect of one of the methods used to track Colony Evaluation for Apis spp. Although it is a method presentation article, the text has an easy and pleasant read. I recommend publishing the article, but with some adjustments that can be found in the text file:

The introduction presents very long paragraphs that make reading tiresome and confusing in a very important piece of the text to understand the importance of the work. Especially at line 50, I suggest that debt in: Line 56 “Orther methods ...”; Line 60 “In additon ...” and Line 67 “A high number…”.

Line 39 - need to insert author / year for Varroa destructor

Line 232 - mention some relevant or pioneering work on the use of this method.

Line 232 - “INRA” change to "National Institute of Agrarian Recherche (INRA)"

Line 275 - remove extra "()" from “(Winston, 1987; Burgett and Burikam (1985))”

Reviewer 3 Report

Overall, I think this approach is a worthwhile addition to the literature and techniques on how to effectively monitor beehive health/strength/resources etc. I do have a few suggestions that would help to clarify the article and justify its claims.

1) Please have the paper read / edited for corrections to English grammar and expression.

2) This might seem a little odd to a person from the biological sciences, but a reader (like this reviewer) who crosses between ecology and computer science may be confused by the appearance of terms such as "an online reference bank for learning and training procedure" (e.g. in the abstract and line 78) without specification of who (or what) is doing the learning / training. I thought at first glance of the abstract that this referred to training data for Machine Learning algorithms. You might add "human learning and training" where this terminology is applied to prevent such confusion in an age where Machine Learning is being applied to exactly the kinds of problem you describe in your paper.

3) I note, in the Abstract, terminology such as "statistical error"... perhaps clarify this by explaining that it is the systematic biases or tendencies you are trying to adjust for. Also, that  by "adjust values recorded" you actually mean not to adjust the values recorded, but to use them to calculate accurate measurements of the hive's statistics. (The former implies editing input data! (That's not a good thing to do.) The latter implies using your understanding of the human biases to make corrections so that your output data is an accurate reflection of the measurements you are trying to make.

4) Keywords: I suggest  some keyword to indicate human estimation/survey of hive properties.

Inserted comment: please do a survey also of standard literature on accounting for biases in field surveys outside of the bee literature. E.g. such things may be conducted to  account for reviewer  biases or tendencies to over- or under-score when assigning scores for scientific papers! But also in trying to account for spatio-temporal biases in species occurrence counts conducted by non-experts (e.g. as in citizen science projects). The same types of problems are rampant in many fields.

5) Line 44 - I recommend that you don't use "optimal" (a very specific technical term that can't be justified here). Perhaps something more like "effective" and "efficient".

6) The arguments from lines 58-63 don't hold weight to me as a rationale for your own work. If they are as you claim (I haven't read the literature you cite here, so I take your remarks at face value) then they are justification for improving the automated imaging techniques but not for avoiding their investigation by resorting to "old style" manual methods. So, instead of the way you phrase the current argument... Simply make an argument that based on previous literature you note that there are currently issues with automatic/image-based approaches. Hence, you have decided instead to focus on improving traditional techniques in your research as that is your preference (or expertise). That would be a legitimate point to make!

Related: why is variability any more of an issue for image-based techniques than for the methods you propose here? (You imply that it is.) Automatic methods can certainly account for variability - probably more easily than can manual methods since the data acquisition method's traits are exactly repeatable!

7) Please do provide sample images for lines 114-115. I don't think it sensible to simply write (as you have done on line 116) that this was, in effect, too difficult. Perhaps showing the images would be instructive - even if only so people can see how hard it is to get good ones!

8) lines 141-145 and 151-155 and 212-214. Please can you explain in the paper, are these values consistent for a user across multiple field survey days? Can you show how this data varies before and after training? How do you account for the benefits of training if, as you suggest, it is done regularly? At  each point mustn't you recompute your bias metrics for every user to keep on top of their current (dynamic, possibly improving with practice) capabilities?

9) Fig  1 refer to parts (a) and (b), or "upper and lower" perhaps.

10) lines 154 - how would the user be made aware of the reason for their over or underestimation? Doesn't your tool just tell them whether or not they over or under estimate?

11) Results and notes on number of observers. I'm unsure how the remark (line 169) about there being 3 observers tallies with Fig. 1 with 29 observers and Fig. 3 with 5 observers. The paper would benefit by organising the figures to correspond with each stage of the process undertaken, being clear in each about numbers of participants and their roles. E.g. Summarise... we conducted N procedures. Procedure 1 was to do X, with 3 participants. Procedure 2 was to do Y with 5  participants. And finally, we did Z using 29 observers. (or something like that). An overview diagram or process placed near the start of your paper would help the reader understand the set of steps you have undertaken without having to figure it out by re-reading earlier sections of the paper to rectify their confusion.

12) What is the "Data collection support tool"? Do you just mean the spreadsheet? If so, best just to call it a spreadsheet to avoid confusing people like me (from computer science and ecological modelling) with a specific "idea" in mind when a "tool" is used - e.g. a special-purpose app. or other interactive software.

13) The use of the technique to survey 15,000 colonies is great! But  some evidence must be presented supporting the claims that it is easy to use (this is a science paper after all). E.g. is there evidence that the training improved staff accuracy in measuring colony structures? (You can't claim this without evidence.) Is there evidence such as from user surveys that staff did find the tool easy to use? (You can't claim this without evidence either!)

14) Line 265... I don't understand why you might need to repeat the estimate for number of honeybees per unit weight... unless the bees' are changing body size or density. Why would they do this? Or is it the weight of the comb that differs between seasons? Can you explain what you mean here? (Sorry if I am ignorant here of something that will be obvious to other readers)

15) What evidence is there that users calibrated themselves? How did they then adjust their performance or in-field survey behaviour? What is the evidence for this?

OVERALL: This is a good, simple, usable idea I think! The paper's claims need to be justified though (or toned down). And the paper would benefit by addressing the concerns above more clearly so that readers are unambiguously informed about what was done, how, and by how many people.

Round 2

Reviewer 1 Report

Authors have efficiently addressed most of my concerns related to their study, they have incorporated many pertinent references related to this subject and clarified the limitation of their proposed tool as well. I believe this study will provide a good addition and contribution to the body knowledge regarding the quantification of honey bee colony performance. Therefore, I am in favor of the publication of this manuscript and see no further scientific issue preventing such decision.